# Integrating Acupuncture into a Dialysis Center

**DOI:** 10.3390/healthcare11101463

**Published:** 2023-05-18

**Authors:** Marta Correia de Carvalho, Pedro Azevedo, Carlos Pires, Jorge Pereira Machado, Manuel Laranjeira, José Nunes de Azevedo

**Affiliations:** 1ICBAS—School of Medicine and Biomedical Sciences, University of Porto, 4050-313 Porto, Portugal; 2TECSAM—Tecnologia e Serviços Médicos SA, 5370-530 Mirandela, Portugal; 3Center for Research in Neuropsychology and Cognitive and Behavioral Intervention (CINEICC), Faculty of Psychology and Educational Sciences, University of Coimbra, 3000-115 Coimbra, Portugal; 4CBSin—Center of BioSciences in Integrative Health, 4000-105 Porto, Portugal; 5INC–Instituto de Neurociências, 4100-141 Porto, Portugal

**Keywords:** integrative medicine, traditional Chinese medicine, acupuncture, hemodialysis, dialysis center, acceptance, feasibility

## Abstract

A growing interest in integrating traditional Chinese medicine (TCM) and conventional medicine (CM) to create a more comprehensive approach to healthcare has been verified. Scientific evidence supports acupuncture as an integrative treatment for specific health conditions. The aim of this study was to assess the acceptance and feasibility by patients and healthcare professionals of integrating acupuncture in a dialysis center. Individuals undergoing hemodialysis (HD) who participated in a patient-assessor-blinded randomized controlled trial that evaluated the effect of acupuncture on functional capacity and quality of life were included. Acceptance was measured by adherence (percentage of patients who completed treatments and dropouts) and patients’ and healthcare professionals’ opinions toward acupuncture (pre- and pro-intervention questionnaires). Feasibility was measured by safety (number of reported adverse events) and effectiveness (changes in functional capacity, peripheral muscle strength, and health-related quality of life scores after treatment). Forty-eight patients were included, and forty-five (93.8%) were analyzed. No adverse events were reported. All patients completed the treatment, and only three patients (6.2%) were lost to the 12-week post-treatment follow-up. The attitudes of patients and health professionals were favorable to acupuncture, namely in relation to its use, degree of discomfort, recommendation to others, and interference with routine care and clinical setting. Integrating acupuncture into a dialysis center seems viable and well-accepted by patients with kidney failure on maintenance HD, doctors and nurses.

## 1. Introduction

Traditional, complementary and integrative medicine has been rising in developing countries with well-established health system structures [1]. According to the World Health Organization (WHO), it encompasses various healthcare practices and therapies, such as traditional Chinese medicine (TCM), that fall outside the scope of conventional medical treatments [2]. The significant increase in the prevalence of traditional and complementary medicine (T&CM) in the last two decades has led regulators to discuss issues related to safety, equity of access, and integration into health systems. As a result, through its first traditional medicine strategy, the WHO has defined global policies with national and regional rules to promote the regulation of products, practices, and practitioner education and training to ensure the rational, safe, and effective use of T&CM [3].

While it is not yet widespread, a few countries have completely incorporated certain T&CM practices into their healthcare systems. For instance, in China, TCM and conventional medicine (CM) are administered together across all levels of healthcare, and both are covered by public and private insurance [2]. 

In addition, research indicates an increasing interest in verifying the combined effect of both T&CM and CM. For example, recent systematic reviews and meta-analyses reported that combining TCM with Western medicine (WM) may be a helpful approach for improving both quality of life and clinical outcomes in breast cancer patients and reducing the incidence of adverse reactions, toxic side effects, traditional and tumor markers [4]; in patients with atopic dermatitis, the combined use of Chinese medicine and WM was found to be more effective than using only WM, enhancing clinical symptoms and quality of life and lowering the recurrence rate [5]. In addition, combining TCM and conventional WM was a viable therapeutic approach for enhancing the clinical symptoms of COVID-19 patients, with no increased adverse events [6].

In the context of Portugal, the designation adopted for these practices is non-conventional therapies (NCT). Their regulation began in 2003 [7] and was recognized in 2019 [8] as one of the bases of the Portuguese health system. NTCs include: acupuncture, phytotherapy, homeopathy, traditional Chinese medicine (TCM), naturopathy, osteopathy, and chiropractic. This action towards regulation resulted, so far, in the attribution of 6311 professional licenses [9,10] which contributed to the client’s safety when using the NCTs with the certainty that an accredited practitioner provided them. Of the NCTs described, only homeopathy has not been awarded professional certification and it is still in the process of regulation. It is worth mentioning that acupuncture and TCM have the highest number of certified professionals following osteopathy [10,11].

Although the Portuguese healthcare system is transitioning towards an integrative approach that incorporates both conventional medicine (CM) and T&CM, NCT is currently not accessible at hospitals and health centers of the National Health System. Its treatments are not covered by health insurance reimbursement [11].

Numerous studies have explored the practicality of incorporating acupuncture into healthcare facilities, and making it accessible to the clinical population, while benefiting jointly from the potential of T&CM and CM [12,13,14,15]. According to the conceptual framework proposed by Zhang et al. (2022), successfully integrating acupuncture in Western hospitals involves converting evidence into clinical practice primarily. In addition, the attitudes of healthcare professionals play a pivotal role in the effective integration of any innovation into healthcare systems [16]. Their attitudes toward acupuncture can vary based on several factors, including their training, cultural background and personal experiences with the therapy. As such, some healthcare professionals may view it with skepticism, while others may be more open to its potential benefits; their attitudes toward acupuncture are complex and multifaceted and may be influenced by various factors [16,17,18,19].

Our prior studies evaluated the impacts of acupuncture on the functional capacity (FC) and health-related quality of life (HRQOL) of patients undergoing hemodialysis (HD), pointing to the possibility of integrating this practice during HD sessions [20,21]. Given the promising results obtained and the limited number of national studies in this field, this paper aims to assess the opinions and perceptions of patients and health professionals regarding the acceptance of incorporating acupuncture as an integrative approach in the context of a Portuguese health services and conventional medicine.

## 2. Methods

### 2.1. Study Design and Participant Criteria

This exploratory study aims to assess patients’ and healthcare professionals’ knowledge, perceptions and opinions about the integration of acupuncture as a complementary therapy for chronic kidney disease (CKD) patients undergoing HD.

Eligible participants were individuals with renal failure on weekly HD treatment for more than three months, male and female, older than eighteen, with a medically stable state, who had participated in a prior patient–assessor randomized trial and had received acupuncture treatment. Individuals with comorbidities such as hypertension and poorly controlled coronary heart disease, angina pectoris, unstable diabetes mellitus control, mental illness, cognitive limitation, or impairment were excluded. In addition, all participants who refused to participate or demonstrated an inability to co-operate with the study procedures were excluded.

Concerning the inclusion criterion, “having received acupuncture treatment”, eligible participants were randomly allocated into the verum and sham acupuncture groups with an allocation ratio of 1:1 through a simple randomization process carried out by an independent researcher, using Microsoft^®^ Excel^®^ for Microsoft 365 MSO to generate randomization series. The participants, the outcome assessor, and the statistician were kept unaware of the allocation of the groups through blinding procedures. However, given the type of intervention, the TCM practitioner was not subjected to blinding. The acupuncture intervention followed a standardized and reproducible protocol designed by the research team, including a comprehensive procedure description according to the revised Standards for Reporting Interventions in Clinical Trials of Acupuncture (STRICTA) 2010 Checklist [22,23]. Briefly, the verum acupuncture (VA) group underwent a total of nine manual acupuncture sessions, using acupoints Taixi (KI3), bilateral, Sanyinjiao (SP6), bilateral, Zusanli (ST36), bilateral, Shenmen (HT7), unilateral in the arm without arteriovenous fistula, and Guan Yuan (CV4), unilateral with manual stimulation. On the other hand, sham acupuncture (SA) received superficial needling (5 mm depth) at non-acupuncture points with no stimulation. Both groups received 25-min needle retention time, and sterilized stainless steel needles (0.25 × 25 mm) were used [22].

Healthcare professionals considered eligible for the study were physicians and nurses working at the dialysis center, providing medical and nursing care related to HD sessions during the administration of acupuncture treatments (from August 2021 to February 2022), and consented to participate. Conversely, those healthcare professionals not present during the acupuncture treatment sessions were excluded from the study. After screening for eligibility, written informed consent was obtained from each participant who agreed to participate.

### 2.2. Data Collection

Acceptance was measured by adherence: first, in terms of the percentage of subjects who completed nine acupuncture sessions and dropped out from the initial assessment to the follow-up assessment, and second by patients’ and health professionals’ views of acupuncture. For this purpose, pre-and post-intervention questionnaires were administered to patients and to health care professionals. 

In order to protect the privacy and confidentiality of study participants, a code was assigned to each questionnaire to ensure data anonymization. 

Throughout the HD session, patients maintained a supine position with restricted use of the upper limbs, specifically the arm with the arteriovenous fistula. Due to this fact and to avoid questioning suggestibility, the questionnaires were administered by an independent researcher who had received previous appropriate training.

The pre-intervention questionnaire (Appendix A) was applied at the baseline and before acupuncture treatment. It included questions about patients’ knowledge of acupuncture, whether they had ever been treated with acupuncture and for what purpose, and whether they would be willing to receive it. After nine acupuncture treatments, the patients answered the post-intervention questionnaire (Appendix A), which contained questions about the discomfort felt when the acupuncture needle was punctured, the discomfort felt during the acupuncture treatment, whether they would recommend it to others, and whether the acupuncture interfered with their HD treatment routine.

A self-administered paper-and-pen format was applied to the doctors and nurses (Appendix A) after the patients completed nine acupuncture treatments, i.e., in the after-treatment time point. It consisted of generic questions about age, gender, profession, and education level. The other questions concerned their knowledge, use, and general opinion about acupuncture and its clinical effectiveness and applicability. They were also asked whether they had ever suggested acupuncture to their patients, whether the treatments provided during the HD session interfered with their care delivery and clinical routine, whether adverse events of acupuncture were observed, and their opinion about the possibility of integrating acupuncture into the dialysis center. In addition to closed questions (“Yes” or “No”), the other questions could be rated according to the Likert 5-point scale. The options were “very positive”, “positive”, “neither positive nor negative”, “negative”, and “very negative”.

Concerning the development of the questionnaire items, clear and concise questions were drawn up to collect the participants’ opinions regarding their knowledge, use, general opinion and recommendation of acupuncture, using simple and appropriate language for the target population. Prior to administering the questionnaires to patients and healthcare professionals, an independent investigator conducted a trial on a small group of patients and healthcare professionals who were unavailable to participate in the study but who met all other inclusion criteria in order to identify problems related to the formulation and understanding of the questions. In this essay, all questions were validated and considered well-formulated.

The feasibility of acupuncture was evaluated by measuring its safety and effectiveness. 

The number of adverse events reported by patients and healthcare professionals was recorded to assess safety. Patients and healthcare professionals were informed of potential adverse events associated with acupuncture before treatment. As acupuncture was administered during the HD session, healthcare professionals were instructed to promptly observe and report any incidents. To facilitate this, a checklist (Appendix B, Figure A1) was made available in the dialysis room for reference during acupuncture treatments. 

The effectiveness of acupuncture was assessed by analyzing the results obtained by the changes between baseline and post-treatment of acupuncture’s effect on FC and HRQOL.

### 2.3. Statistical Analysis

Patients’ sociodemographic and clinical characteristics, and the answers to the questionnaire were described, by group, through frequencies (n and %). Minimum, maximum, mean, and standard deviation were used for age. Fishers’ Exact Test and the two sample *t*-test were used for group comparisons. Normality of data was assessed and validated with the Shapiro–Wilk test. The same descriptive measures were used for healthcare professionals’ sociodemographic characteristics and their answers to the questionnaire. 

A significance level of 5% was considered for the statistical tests. Statistical analysis was performed with IBM SPSS Statistics software-version 27.0 [24].

## 3. Results

A total of forty-five patients were included, aged between 57 and 91, with a mean age of 71.9 (SD = 6.8). Most were men (62.2%), lived in rural areas (62.2%), with low education levels (≤6 years of schooling) (93.3%), and retired (86.7%). In terms of clinical features, diabetes mellitus (44.4%) was the leading causes for CKD; most underwent HD between 10 and 120 months (86.7%), with arteriovenous fistula (88.9%) being the most common method of vascular access, as shown in Table 1. The group of patients who received verum and sham acupuncture did not differ significantly (*p* > 0.05) regarding sociodemographic or clinical characteristics.

Regarding the sociodemographic profile of the participating health professionals (Table 2), 41.7% were female and 58.3% were male, with an average age of 51.2 (SD = 13.7). The majority held a bachelor’s degree (83.3%), with 41.7% being doctors and 58.3% being nurses. Among the doctors, three (60.0%) were specialists in general practice/family medicine and two (40.0%) in nephrology. The three nurses who were specialists were in the areas of medical–surgical, rehabilitation and public health.

### 3.1. Acceptance

#### 3.1.1. Adherence

All participants completed acupuncture treatment throughout the study period, indicating a 100% adherence rate. However, three dropouts occurred between the post-treatment and follow-up assessment due to kidney transplantation, hospitalization and death, resulting in a 6.6% dropout rate (Table 3). 

#### 3.1.2. Patients’ and Healthcare Professionals’ Attitudes toward Acupuncture

From the pre-intervention questionnaire data (Table 4), out of the 45 participants, more than half (55.6%) had no knowledge about acupuncture. Concerning acupuncture use, only one participant (2.2%) had had acupuncture treatment for osteoarticular pain more than three months before. However, all were receptive to receiving acupuncture treatment (100%). No statistically significant differences were found between the groups (*p* > 0.05). 

From the post-intervention questionnaire data analysis (Table 5), regarding the degree of discomfort felt when puncturing the acupuncture needle, 31.1% answered “uncomfortable,” 48.9% “neither comfortable nor uncomfortable,” and 20% “comfortable.” Regarding the degree of discomfort felt during acupuncture treatment, 17.8% answered “neither comfortable nor uncomfortable,” 35.6% answered “comfortable,” and 46.7% “very comfortable.” Furthermore, the majority (91.1%) would recommend acupuncture treatment to others. All participants considered that acupuncture treatments did not interfere with the hemodialysis routine. There were no statistically significant differences between the groups (*p* > 0.05).

Regarding the results from health professionals’ questionnaire (Table 6), all participants were aware of what acupuncture entailed. Furthermore, 33.3% of the respondents had received acupuncture treatment for osteoarticular pain and/or muscle pain for more than three months. In addition, all the health professionals would recommend acupuncture treatment to others, and 91.7% expressed openness to receiving acupuncture treatment themselves.

In terms of general opinion about acupuncture, 16.7% answered “neither negative nor positive,” 66.7% answered “positive,” and 16.7% “very positive.” As for acupuncture’s efficacy and clinical applicability, 8.3% evidenced a neutral stance opinion, while 75% evidenced “positive” and 16.7% “very positive.” All considered that the acupuncture treatments provided to patients did not interfere with routine hemodialysis. 

Only 8.3% of the respondents expressed a neutral stance about integrating acupuncture in patient care during hemodialysis, while the rest held “positive” (33.3%) or “very positive” (58.3%) opinions.

### 3.2. Feasibility

#### 3.2.1. Safety

Throughout the trial period, patients or health professionals observed or reported no acupuncture-related adverse events, suggesting the safety of using acupuncture during HD sessions.

#### 3.2.2. Effectiveness

The effectiveness of acupuncture was assessed by scrutinizing the overall results described in a previous RCT [20,21]. The group that received verum acupuncture (VA) demonstrated better results, compared to the group that received sham (SA) acupuncture and a control group. Specifically, they walked a greater distance, exhibited increased peripheral muscle strength in their lower limbs and handgrip strength. Additionally, they achieved higher scores in HRQOL, as evidenced by an increase in KDQOL-SFTM 1.3 Physical component summary (PCS) score, which includes physical functioning, role functioning/physical, bodily pain, general health, vitality, and social functioning, as well as in a mental component summary (MCS) score, overall health and specific areas targeted by kidney disease (symptom/problem list, effects of kidney disease, cognitive function, sleep).

The participant blinding was assessed by asking patients which type of acupuncture treatment they believed they received. Most patients in the VA and SA groups were uncertain about their treatment allocation. However, 31% of patients believed they received verum acupuncture, with similar percentages in both groups. None of the patients believed they received sham acupuncture. The calculated blinding index (0.84–95% CI: 0.78–0.91) indicated successful participant blinding.

## 4. Discussion

Integrating traditional and complementary medicine, such as acupuncture, into public or private conventional healthcare services is contingent on multiple considerations. In this regard, we assessed its acceptance and feasibility to determine the possibility of incorporating acupuncture as an adjuvant and complementary therapy in a dialysis center.

The adherence indicators and the patient and health professional opinion questionnaire were used to check acceptance. A complete adherence rate was found from baseline to post-treatment evaluation, constituting an important indicator of commitment to acupuncture treatment. 

The literature review reports a low frequency of traditional and complementary medicines for individuals in maintenance HD [25]. The same was found in our sample of patients, where only one had performed acupuncture treatment for osteoarticular pain before they participated in this study.

Regarding the patient’s experience of receiving acupuncture, although some discomfort has been reported when puncturing the acupuncture needle, this did not occur during treatment. Acupuncture was well accepted by the patients, as evidenced by their high level of recommendation for the treatment to others and the fact that it did not disrupt their HD routine.

The optimal implementation of innovative health and patient-centered care strategies relies not only on patients’ knowledge and beliefs but also the attitudes of healthcare professionals [16]. The healthcare professionals’ survey outcomes indicated that doctors and nurses from TECSAM’s dialysis center revealed knowledge about acupuncture, and the majority expressed their willingness to receive acupuncture treatment. However, while a few held a neutral general opinion of acupuncture and its efficacy and clinical applicability, most had an optimistic perspective. Doctors and nurses also unanimously agreed that acupuncture treatments did not interfere with the dialysis routine. Most of them held a favorable opinion regarding including acupuncture as a part of regular care for patients undergoing HD. It appears that the level of knowledge of health professionals on the subject may have influenced the results obtained, considering that the dialysis center where the study was conducted is known to adopt an integrative approach in the care of patients undergoing HD. In addition, health professionals’ direct observation of acupuncture treatments during HD may have contributed to this fact. Finally, the authors highlight the importance of providing information and training to health professionals on the scientific evidence of the application of acupuncture in clinical practice to broaden the acceptance of these complementary therapeutic approaches.

Unfortunately, given the lack and heterogeneity of studies evaluating health professionals’ attitudes toward acupuncture and other T&CM practices, it was not possible to obtain comparative results. Despite this limitation, some studies have reported similar results. The study conducted by Zhang et al. (2022) found that doctors and nurses held positive viewpoints on acupuncture and acupressure use in perioperative care [26]; Shao et al. (2005) reported broadly positive attitudes of general practitioners and hospital doctors toward traditional acupuncture [18]; Lipman et al. (2003) study findings, suggested that the majority of general practitioners were supportive of making acupuncture readily accessible within the National Health Service (NHS) [19]. Recent reviews have also shown positive attitudes toward acupuncture among healthcare professionals [27].

Concerning the feasibility, assessed in terms of safety and efficacy, no adverse effects acupuncture-related were reported or recorded. Furthermore, its effectiveness in improving FC and HRQOL has been verified [20,21]. Despite the limited number of randomized clinical trials examining the safety and efficacy of acupuncture in CKD patients receiving maintenance HD, these parameters have been extensively investigated and established across diverse medical conditions [28,29,30,31,32,33,34]. Nevertheless, the results of a recent review and meta-analysis indicate that acupuncture may be a safe and effective therapeutic alternative for uremic pruritus in patients undergoing HD. Additionally, the combination of acupuncture and HD effectively alleviated uremic pruritus symptoms compared to HD alone [35].

In line with what these studies point out, the results of the present study indicate that acupuncture is safe and effective for patients receiving maintenance HD.

As the limitation of the study, it is worth noting the small sample size of healthcare professionals and the fact that only the questionnaire’s content and construct validity assessment was conducted without a pilot study to evaluate its validity and reliability.

Based on the findings obtained from this study, acupuncture should be considered and incorporated as an integrative approach in CM health services. In addition to the patient’s and healthcare professionals’ acceptance, it is also essential to carefully consider the views of healthcare management and administrative boards and the acceptance of other stakeholders.

Although our results are promising, future larger RCTs to validate the safety and effectiveness of acupuncture on patients undergoing HD are required. In addition, further research should involve cost-benefit analyses, comparing outcomes and acupuncture treatment costs with other conventional therapies (e.g., physiotherapy) on patient functional capacity. Finally, it will be necessary to create a conceptual framework to address logistical issues associated with integrating acupuncture into conventional care and verify its clinical applicability to other populations or settings.

## 5. Conclusions

In conclusion, the acceptance and feasibility assessment findings seem to support the integration of acupuncture as a viable and beneficial complementary treatment option for managing the symptom burden and enhancing the quality of life of patients undergoing maintenance HD. Extended future research is required.

## Figures and Tables

**Table 1 healthcare-11-01463-t001:** Patients’ sociodemographic and clinical characteristics.

		VA Group (n = 23)	SA Group (n = 22)	Total (N = 45)	*p*-Value
**Sociodemographic**					
Age	Minimum–Maximum	60–84	57–91	57–91	*0.496 ^(2)^*
	Mean (SD)	71.2 (5.1)	72.6 (8.3)	71.9 (6.8)	
		n (%)	n (%)	n (%)	
Gender	Female	9 (39.1)	8 (36.4)	17 (37.8)	*1.000 ^(1)^*
	Male	14 (60.9)	14 (63.6)	28 (62.2)	
Residence	Urban	9 (39.1)	8 (36.4)	17 (37.8)	*1.000 ^(1)^*
	Rural	14 (60.9)	14 (63.6)	28 (62.2)	
Education level	No literacy	0 (0.0)	2 (9.1)	2 (4.4)	*1.000 ^(1)^*
	1° Cycle (4 years)	20 (87.0)	17 (77.3)	37 (82.2)	
	2° Cycle (6 years)	2 (8.7)	1 (4.5)	3 (6.7)	
	High school (12 years)	1 (4.3)	2 (9.1)	3 (6.7)	
Professional status	Employed	0 (0.0)	2 (9.1)	2 (4.4)	*0.782 ^(1)^*
	Self-employed	2 (8.7)	0 (0.0)	2 (4.4)	
	Unemployed	1 (4.3)	1 (4.5)	2 (4.4)	
	Retired	20 (87.0)	19 (86.4)	39 (86.7)	
**Clinical**		n (%)	n (%)	n (%)	
CKD causes	Diabetes mellitus	10 (43.5)	10 (45.5)	20 (44.4)	*0.202 ^(1)^*
	Chronic rejection	1 (4.3)	4 (18.2)	5 (11.1)	
	Hypertensive nephropathy	3 (13.0)	0 (0.0)	3 (6.7)	
	High blood pressure	0 (0.0)	1 (4.5)	1 (2.2)	
	Glomerulonephritis	1 (4.3)	0 (0.0)	1 (2.2)	
	Interstitial tubular necrosis	0 (0.0)	2 (9.1)	2 (4.4)	
	Other	3 (13.0)	1 (4.5)	4 (8.9)	
	Unknown	5 (21.7)	4 (18.2)	9 (20.0)	
Hemodialysis time	<12 months	2 (8.7)	0 (0.0)	2 (4.4)	*0.544 ^(1)^*
	12 to 120 months	19 (82.6)	20 (90.9)	39 (86.7)	
	>120 months	2 (8.7)	2 (9.1)	4 (8.9)	
Vascular access	Arteriovenous fistula	22 (95.7)	18 (81.8)	40 (88.9)	*0.187 ^(1)^*
	Central venous catheter	1 (4.3)	4 (18.2)	5 (11.1)	

VA = Verum Acupuncture; SA = Sham Acupuncture; *^(1)^* significance value of Fisher’s Exact Test; *^(2)^* significance value of two sample *t*-test.

**Table 2 healthcare-11-01463-t002:** Healthcare professionals’ sociodemographic characteristics (N = 12).

		n (%)
Gender	Female	5 (41.7)
	Male	7 (58.3)
Age	Minimum–Maximum	23–73
	Mean (SD)	51.2 (13.7)
Education level	Bachelor’s Degree	10 (83.3)
	Master’s Degree	2 (16.7)
Profession	Doctor	5 (41.7)
	Nurse	7 (58.3)
Medical specialty	General Practice/Family Medicine	3 (25.0)
(Doctors: N = 5)	Nephrology	2 (16.7)
Nursing specialty	Medical-surgical	1 (14.3)
	Rehabilitation	1 (14.3)
	Public health	1 (14.3)
(Nurses: N = 7)	None	4 (57.1)

**Table 3 healthcare-11-01463-t003:** Number of participants at each evaluation time point.

	VA Group (n)	SA Group (n)	Total (n/%)	*p*-Value *^(1)^*
Baseline assessment	24	24	48 (100)	
Treatment	24	24	48 (100)	
Post-treatment assessment	24	24	48 (100)	
12-week follow-up assessment	23	22	45 (93.4)	
Lost in follow-up/Dropout	1	2	3 (6.6)	
Reason	Kidney transplantation	Hospitalization, death		
Analyzed-n (%)	23 (95.8%)	22 (91.7%)	45 (93.4%)	*1.000*

VA = Verum Acupuncture; SA = Sham Acupuncture; *^(1)^* significance value of Fisher’s Exact Test.

**Table 4 healthcare-11-01463-t004:** Pre-intervention patient’ questionnaire.

		VA Group (n = 23)	SA Group (n = 22)	Total (N = 45)	*p*-Value *^(1)^*
		n (%)	n (%)	n (%)	
Do you know what Acupuncture is?	No	11 (47.8)	14 (63.6)	25 (55.6)	*0.373*
	Yes	12 (52.2)	8 (36.4)	20 (44.4)	
Have you received any Acupuncture treatment before your participation in this study?	No	23 (100.0)	21 (95.5)	44 (97.8)	*0.489*
	Yes	0 (0.0)	1 (4.5)	1 (2.2)	
If so, how long ago has it been?			>3 months		*-*
And for what purpose?			Osteoarticular pain		*-*
Are you willing to receive an Acupuncture treatment?	No	0 (0.0)	0 (0.0)	0 (0.0)	*-*
	Yes	23 (100.0)	22 (100.0)	45 (100.0)	

VA = Verum Acupuncture; SA = Sham Acupuncture; *^(1)^* significance value of Fisher’s Exact Test.

**Table 5 healthcare-11-01463-t005:** Post-intervention patient’ questionnaire.

		VA Group (n = 23)	SA Group (n = 22)	Total (N = 45)	*p*-Value *^(1)^*
		n (%)	n (%)	n (%)	
Degree of discomfort felt when puncturing the acupuncture needle.	Totally uncomfortable	-	-	-	*0.858*
	Uncomfortable	8 (34.8)	6 (27.3)	14 (31.1)	
	Neither comfortable, nor uncomfortable	11 (47.8)	11 (50.0)	22 (48.9)	
	Comfortable	4 (17.4)	5 (22.7)	9 (20.0)	
	Totally comfortable	-	-	-	
Degree of discomfort during acupuncture treatment.	Totally uncomfortable	-	-	-	*0.488*
	Uncomfortable	-	-	-	
	Neither comfortable, nor uncomfortable	3 (13.0)	5 (22.7)	8 (17.8)	
	Comfortable	10 (43.5)	6 (27.3)	16 (35.6)	
	Totally comfortable	10 (43.5)	11 (50.0)	21 (46.7)	
Would you recommend acupuncture treatment to others?	No	2 (8.7)	2 (9.1)	4 (8.9)	*1.000*
	Yes	21 (91.3)	20 (90.9)	41 (91.1)	
In your opinion, did the Acupuncture treatments interfere with your hemodialysis routine?	No	23 (100.0)	22 (100.0)	45 (100.0)	*-*
	Yes	-	-	-	

VA = Verum Acupuncture; SA = Sham Acupuncture; *^(1)^* significance value of Fisher’s Exact Test.

**Table 6 healthcare-11-01463-t006:** Healthcare professionals’ questionnaire (N = 12).

		n (%)
Do you know what Acupuncture is?	No	-
	Yes	12 (100.0)
Have you ever received any acupuncture treatment?	No	8 (66.7)
	Yes	4 (33.3)
If so, how long ago has it been?	>3 months	4 (100.0)
And for what purpose?	Osteoarticular pain	2 (50.0)
	Muscle pain	3 (75.0)
Would you recommend acupuncture treatment to others?	No	-
	Yes	12 (100.0)
Would you be receptive to receiving acupuncture treatment?	No	1 (8.3)
	Yes	11 (91.7)
What is your general opinion on Acupuncture?	Very negative	-
	Negative	-
	Neither negative nor positive	2 (16.7)
	Positive	8 (66.7)
	Very positive	2 (16.7)
What is your opinion on the efficacy and clinical applicability of Acupuncture?	Very negative	-
	Negative	-
	Neither negative nor positive	1 (8.3)
	Positive	9 (75.0)
	Very positive	2 (16.7)
Have you ever suggested acupuncture to a patient?	No	4 (33.3)
	Yes	8 (66.7)
In your opinion, did the acupuncture treatments provided to patients interfere with the hemodialysis routine?	No	12 (100.0)
	Yes	-
What is your opinion on the integration of acupuncture into the care of patients on hemodialysis?	Very negative	-
	Negative	-
	Neither negative nor positive	1 (8.3)
	Positive	4 (33.3)
	Very positive	7 (58.3)

## Data Availability

Not applicable.

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
