# Peer review of "Integrating Acupuncture into a Dialysis Center"

_healthcare, 2023, doi:10.3390/healthcare11101463_

Round 1

Reviewer 1 Report

As acupuncture therapy has been used for patients in more countries than exepcted, it would be positve and contributable for the integrative medicine. However, some concerns has to be suggested as follows;

 1) Overall, in this manuscripts, It is necessary to describe in more detail so that the efforts of the researchers can be clearly expressed well to apply acupuncture theray in healthcare systems as integrative medicine.

 2) In Manuscripts, Fishers’ Exact Test and the Student’s T-Test were used for group comparisons and Table 1 - Table 5

âžœ In Table1 and Table5, it has to be confirmed if Fishers’ Exact Test is    appropriate even it’s not 2*2 table

➜ two sample t-test is more appropriate description than Student t-test

âžœ As the number of VA group is 23 and the number of SA group is 22 in Table1, it’s asking if the normality test was performed and satisfied to apply two sample t-test

➜ In Table 1- Table5, for all characteristis except age, % % % ..are repeatedly expressed. it's suggested to find the way to express more clearly and consistently.

 3) For the patients, it’s just considered CKD causes, HDtime, and Vascular access. it’s wondering there’s no other clinical characteristics that should be considered for HD including other convetional treatments of participants and underline diseases. In addtion, inclusion and exclusion of patients has to be describle more in detail.

 4) In Healthcare Professionals´ QUESTIONNAIRE ( it’s given as supplementary) and Table 6 even the Table4, it’s questionable to make the treatment time be 3month. If there’s some guideline and concensus, it’s better to describle in the manuscript.

 5) It’s suggested to describe cllinical research diagram including RCT procedure and the frequency of acupuncture treatment for 12 weeks to make the process well known and understood if it’s blinded or not.

 6) It’s wondering if the questionaire is confirmed one or only developed for this research. If it’s developed only for this research, the description for the process to include all items would be better to improve the confidence of this research.

7) In manuscripts, there's no adverse effects reported. If so,  the monitoring has to be mentioned for the clear process. 

According to authors,  it's the first paper to describe the viability of acupuncture practice in a private hemodialysis center in Portugal. Yes, It's fine to get the small study group even healthcare professionals.  To make this paper more contributable in Integrative medicine area, above concerns and suggestions would be considered to step forward .  

Thank you.

Reviewer 2 Report

The study is well-designed, but valuable ideas have been used in previous publications to which the authors refer.

This study has certain inconsistencies that represent serious methodological obstacles in answering the hypothesis given in the paper's title.

The reliability of the answers of elderly patients, with a low educational profile, from rural areas, and at the same time without verified data on comorbidities related to their cognition is questionable. 

All this seriously compromises the reliability of the answers obtained and the study in general.

I think the implementation of traditional Chinese medicine within conventional/Western medicine (based on evidence) is a very delicate area that requests extensive study protocols and long-term follow-up at various levels, and according to that, positive reviews, which are given lightly in studies like this, can guide to the wrong direction.

This study has certain shortcomings.
In the Methodology section, certain data is missing needed to provide a clearer insight into the study design.
First, in what way is one private hospital considered representative, and how do authors explain such extraordinary affection of health workers towards acupuncture?
Secondly, there is a lack of data on the comorbidities (dementia, mental disorders...) of the oldest respondents and their distribution by age. Also, how did the respondents fill out the questionnaires, and were there difficulties for the oldest? Given that these are old, sick, and uneducated patients, did they understand the questions asked, and if they answered in an interview, how was the questioning suggestibility avoided? How the validity and reliability of the research Instruments were explicitly confirmed for the specific age of respondents.

The discussion is written so that it almost entirely repeats the sentences from the Results section with minimal supplementary references or any scientific assumptions that would explain certain doubts about the referred results. Authors mainly direct to auto-citations of publications from the same study they published in this journal (Healthcare) most recently.

Reviewer 3 Report

In this manuscript, the authors assessed the feasibility and acceptability of integrating acupuncture by patients and healthcare professionals in a dialysis center. Patients undergoing hemodialysis were included who have participated in a patient-assessor blinded randomized controlled trial (RCT) that evaluated the effect of acupuncture on functional capacity and quality of life. They found that the attitudes of patients and health professionals were favorable to acupuncture, namely in relation to its use, degree of discomfort, recommendation to others, and interference with routine care and clinical setting. Verum acupuncture increased the walk distance, the lower limbs and hand-grip strength, and overall quality of life. Integrating acupuncture into a dialysis center seems feasible and well accepted by doctors and nurses and patients with kidney failure on maintenance of HD.

A few issues discussed in the following:

1.     In the methods section, please provide the detail of inclusion and exclusion criteria of eligible dialysis patients participants.

2.     Please provide the detail of inclusion and exclusion criteria of healthcare providers.

3.     Please provide the method of randomization of participants to verum and sham acupuncture groups.

4.     What is the VA and SA treatment parameters, including the acupuncture point selection, needle stimulation type (manual, electric), duration of the treatment and frequency of the treatment, etc.

Round 2

Reviewer 2 Report

Kudos to the authors for very clear answers adopted suggestions and made corrections.

The paper now presents the research idea much more clearly and enables a better understanding of the hypothesis. This increased the quality and scientific contribution of the obtained results